



# North Atlantic response to a quasi-realistic Greenland meltwater forcing in eddy-rich EC-Earth3P-VHR hosing simulations

Eneko Martin-Martinez[1,2], Eduardo Moreno-Chamarro[3], Fraser William Goldsworth[3], Jin-Song von Storch[3], Cristina Arumí-Planas[1], Daria Kuznetsova[1], Saskia Loosveldt-Tomas[1], Pierre-Antoine Bretonnière[1], and Pablo Ortega[1]

[1]Barcelona Supercomputing Center (BSC), Barcelona, Spain
[2]Departament de Dinàmica de la Terra i l'Oceà, Facultat de Ciències de la Terra, Universitat de Barcelona (UB), Barcelona, Spain
[3]Max Planck Institute for Meteorology, Hamburg, Germany

**Correspondence:** Eneko Martin-Martinez (eneko.martin@bsc.es)

**Abstract.**

The vast majority of studies examining the impact of freshwater from ice sheet melting on the Atlantic Meridional Overturning Circulation (AMOC) use climate models that cannot resolve mesoscale ocean processes and do not include an accurate spatio-temporal distribution of the freshwater forcing. These two factors critically affect the nature of the AMOC response. Our study fills that gap with a set of three hosing experiments performed with the global configurations of the eddy-rich climate model EC-Earth3P-VHR. The model is forced for 21 years with a spatial and monthly distribution of Greenland meltwater fluxes derived from observations, equal to 0.04 Sv on an annual average.

Within the first year, we observe a response of reduced salinity in the Greenland and Labrador currents. This is accompanied by an acceleration and a cooling along the currents that lead to a rapid weakening of the AMOC at subpolar latitudes. Around year 7, deep mixing in the Labrador Sea begins to weaken due to as freshwater anomalies accumulate through lateral exchanges with the boundary currents. This shallowing of the mixed layer further weakens the AMOC, resulting in a stronger reduction that reaches also the subtropical latitudes. By the end of the simulation, the AMOC has weakened by almost 3 Sv at subppolar latitudes (i.e. a decrease of around 20 %), with an average relative decrease of 10 % for the whole Northern Hemisphere. The reduction in the AMOC is strong enough for some global climate impacts to emerge, such as the "bipolar seesaw" temperature response.

## 1 Introduction

Anthropogenic global warming is forcing the Greenland Ice Sheet to melt, injecting a large amount of freshwater into the surrounding ocean (Bamber et al., 2018; Pattyn et al., 2018). The associated changes in salinity modify the surface density, which in turn can affect the local circulation. Strong freshwater forcing could also significantly reduce deep water formation in the Subpolar North Atlantic (SPNA), which is linked to the weakening in the Atlantic Meridional Overturning Circulation (AMOC) (Jackson and Wood, 2018; Martin et al., 2022; Bellomo et al., 2023; Jackson et al., 2023; Martin and Biastoch, 2023;





Ma et al., 2024; Wei and Zhang, 2024). However, it is still largely uncertain whether the projected AMOC reduction will result in its complete shutdown (van Westen et al., 2024; Baker et al., 2025; Loriani et al., 2025; Winkelmann et al., 2025).

A moderate AMOC weakening could also have significant climate impacts. By reducing the northward heat transport,
AMOC reductions induce a "bipolar seesaw" temperature response, with pronounced cooling across the Northern Hemisphere (NH) balanced by modest warming in the Southern Hemisphere (SH) (Jackson et al., 2015; Liu et al., 2020; van Westen et al., 2024; Diamond et al., 2025). This anomalous interhemispheric temperature gradient drives a southward displacement of the Intertropical Convergence Zone (ITCZ) (Jackson et al., 2015; Liu et al., 2020; Orihuela-Pinto et al., 2022; Bellomo et al., 2023; Ma et al., 2024), a reduction in precipitation over NH mid-latitudes (Jackson et al., 2015; Bellomo et al., 2023)
and a strengthening of the Pacific Walker circulation (Orihuela-Pinto et al., 2022). The associated changes in atmospheric dynamics can lead to complex regional impacts: experiments with a 60 % reduction of the AMOC in eddy-parameterised climate models show that while the intensified wintertime NH jet stream reduces the frequency of prolonged cold spells over much of Europe (Meccia et al., 2023), the summer weakening of the jet stream increases the frequency of Ural atmospheric blocking, contributing to more frequent heatwaves in Eastern Europe (Meccia et al., 2025). In polar regions, reduced northward
heat transport leads to a slowdown in Arctic sea ice loss (Liu et al., 2020), while concurrent increased southward ocean heat transport in the Southern Ocean causes multidecadal local surface warming and sea-ice loss (Diamond et al., 2025). Crucially, AMOC weakening also diminishes the ocean's ability to transport carbon-rich waters to the deep ocean, thereby reducing ocean carbon uptake (Schaumann and Alastrué de Asenjo, 2025).

Freshwater hosing experiments aim to isolate the impact of ice sheet melted waters on the climate system using idealized
model experiments (Devilliers et al., 2021; Martin et al., 2022; Swingedouw et al., 2022; Bellomo et al., 2023; Jackson et al., 2023; Martin and Biastoch, 2023; Meccia et al., 2023; Schiller-Weiss et al., 2023; Devilliers et al., 2024; Ma et al., 2024; Oltmanns et al., 2024; Schiller-Weiss et al., 2024; Wei and Zhang, 2024; Diamond et al., 2025; Meccia et al., 2025). These experiments have also become a popular technique for studying the potential recovery of the AMOC after a strong weakening. In particular, the North Atlantic Hosing Model Intercomparison Project (NAHosMIP) aims to study the stability of the
AMOC using coupled climate models with resolutions ranging from eddy-parameterised (approximately 100 km of horizontal resolution) to eddy-permitting (approximately 25 km of horizontal resolution) (Jackson et al., 2023). NAHosMIP proposes two different protocols for injecting freshwaters into the North Atlantic: a uniform widespread injection from 50° N to the Bering Strait and a more realistic injection localized around Greenland with an exponential decay up to 300 km from the coast. Their main objective is to study AMOC recovery in Coupled Model Intercomparison Project phase 6 (CMIP6) models
by shutting down the forcing after a set number of years (e.g. 50 years of hosing and 100 years of recovery). From a total of eight models, half recovers from a weak AMOC state, while the other half remains in the weak state. They also conclude that these differences in behaviour can not be explained by model resolution (within the resolutions considered) or mean climate state. However, NAHosMIP does not include eddy-rich simulations, which may have a different response due to the impact of mesoscale eddies and properly resolved boundary currents.

Resolving mesoscale ocean eddies reduce biases in the mean state of the ocean, particularly with regard to salinity in the SPNA (Jüling et al., 2021; Frigola et al., 2025). Mesoscale eddies and better resolved boundary currents also improve AMOC





pathways and its variability (Hirschi et al., 2020). Therefore, eddy-rich models are ideal to conduct a more in depth investigation of how a realistic meltwater forcing would affect the AMOC and its impacts in the coming years. Few studies have already shown the impact of resolving eddies in the distribution of freshwater in the SPNA (Böning et al., 2016; Martin and Biastoch, 2023). In order to make the most of the finer resolution, a protocol with a greater focus on the imminent transient response than on long-term equilibrium is needed. An overly idealized Greenland hosing configuration can result in an unrealistic distribution of injected freshwater (Goldsworth, 2026). So far, few studies have used eddy-rich simulations with more realistic hosing simulations, distributing the meltwater based on observations. This reproduces the spatial distribution and seasonality of the freshwater. However, these simulations with heterogeneous distributions are only eddy-rich in the North Atlantic (Martin and Biastoch, 2023; Schiller-Weiss et al., 2023, 2024).

This paper presents a set of Greenland freshwater hosing experiments conducted with EC-Earth3P-VHR, an eddy-rich model developed for the High-Resolution Model Intercomparison Project (HighResMIP; Haarsma et al., 2016). To our knowledge, this is the first time a coupled global climate model with an eddy-rich ocean and a high resolution atmosphere has been used with such a realistic temporal and spatial distribution of meltwater fluxes around Greenland. Section 2 describes the methods, including the model configuration and experimental setup (Section 2.1) and significance tests considered in the analyses (Section 2.2). Section 3 describes the results and includes subsection 3.1 that explores the response of the AMOC to the freshwater hosing, subsection 3.2 that characterises the wider ocean changes across the North Atlantic, subsection 3.3 that investigates how these responses evolve over time and relate to one another, and section 3.4 that examines the atmospheric impacts at the global scale. Section 4 summarizes the conclusions and discusses remaining open questions.

## 2  Methodology

### 2.1  Experimental set-up

We use the global coupled climate model EC-Earth3P-VHR (Moreno-Chamarro et al., 2025), an eddy-rich version of the model specifically developed within the PRIMAVERA project to contribute to the first phase of HighResMIP (Haarsma et al., 2016), as part of the CMIP6 initiative. This model version uses the atmospheric IFS cy36r4 model, the ocean NEMO model in its version 3.6, and the sea ice LIM3 model. This eddy-rich configuration of the model has an approximate grid spacing of about 8 km in the ocean (ORCA12 grid) and about 16 km in the atmosphere (T1279 grid), both at mid-latitudes.

We leverage the high-resolution of the model to propose a quasi-realistic approach to freshwater hosing. We apply the freshwater hosing according to a spatial and temporal distribution based on observations from Bamber et al. (2018), see Fig. 1. We inject a constant freshwater flux of 0.1 Sv (1 Sv = $10^6$ m$^3$ s$^{-1}$) averaged during the melting months (May-September according to Bamber et al. (2018)). This is equivalent to an annual average of 0.04 Sv or a total of 1322 km$^3$yr$^{-1}$. This quantity is close to estimations for the values from 2013–2016 (Bamber et al., 2018) and is slightly smaller than the idealized 0.05 Sv on annual average used in other similar studies (Martin et al., 2022; Martin and Biastoch, 2023; Wei and Zhang, 2024).

We include the freshwater as an additional term to model river runoff (Fig. B1 shows the extra runoff input to the model) that is evenly and instantaneously distributed along several ocean coastal points connected to each hydrological basin in the vicinity





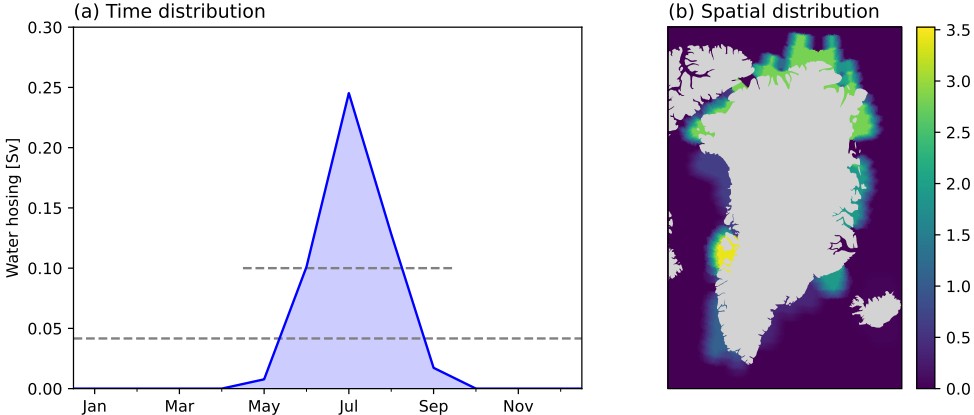

**Figure 1.** (a) Total monthly freshwater forcing time distribution and (b) May-September averaged freshwater forcing runoff spatial distribution. Values computed from the averaged difference in the runoff (model output) for the first year of each hosing member. The dashed lines in (a) indicate the annual average and the average over the five months of forcing. This figure corresponds to the smoothed and interpolated input data, shown in Fig. B1, after being smoothed by the model coupler.

of the major river outlets, as well as vertically to avoid numerical and physical problems (Gurvan et al., 2017), at 0 psu and local seawater temperature. This implementation uses the river runoff routine available in the CMIP6 EC-Earth model (more details in Döscher et al. (2022)). However, we use an improved version of the runoff mapper, in which Greenland is partitioned into six sub-drainage basins corresponding to its main glaciological drainage basins, instead of the single Greenland-wide drainage basin in the original implementation. We do not apply compensation for the salinity, as this does not happen in the

real world; therefore, the global mean salinity is not conserved.

We run three ensemble members of 21 years, branching off from three different initial states of the HighResMIP 1950-control Moreno-Chamarro et al. (2025) representing a weak, a moderate and a strong AMOC states, respectively; the start dates are additionally chosen to minimise overlap between experiments, ensuring there was at least a 15-year gap between them. To avoid the effects of a strong initial model drift, the first 30 years of the control are also discarded when selecting the

start dates. The control uses perpetual radiative forcing conditions from 1950 and will be used as the benchmark experiment (control) to isolate the effects of the hosing. The three hosing runs use the same freshwater flux and constant 1950-forcing. Martin-Martinez et al. (2025) gives a detailed analysis of the mean state and internal variability of the SPNA and AMOC in the control experiment.

## 2.2 Results evaluation

All the scientific analyses are performed using well-established open-source scientific programming languages and tools. Most of the analyses are performed directly with Python and ESMValTool v2.12.0 (Righi et al., 2020; Andela et al., 2025a, b), a





software package specifically created to facilitate a rigorous evaluation of CMIP simulation outputs that is especially useful to compare multiple models with observational datasets.

The anomalies between the hosing and control experiments are computed by matching the years between both experiments after the initialization. This is intended to remove any remaining model drift and align the initial internal climate variability. The statistical tests are tailored to the relatively small ensemble size, with only three members.

Statistical significance when computing time averages (e.g. the last 10 years of a variable) is assessed with a bootstrap method, using annual (or seasonal/monthly) data as the sample elements. As we have a 3-member ensemble, this way the sample size will be three times the number of averaged years; for example, 30 elements are used for an average over the last 10 years. We generate 1000 bootstrapped samples from the previous sample of anomalies to check if the difference is significant at the 95 % confidence level.

In time series and Hovmöller diagrams, bootstrap cannot be applied with the same method, as we would only be able to use the three elements as the original sample. Therefore, we use a consistency test, and check if the anomaly sign is consistent across the three members. However, as alignment between the three members may frequently occur by chance, we only consider a signal to be coherent when it persists over time for at least three years and can be physically explained.

## 3 Results

### 3.1 Meltwater impacts on the AMOC

We first examine the volume overturning streamfunction to determine how and when the freshwater hosing impacts the AMOC. Figure 2a shows the time evolution of the maximum overturning in depth space at 33.8° N, that is the latitude where the AMOC exhibits the strongest response (Figure 3a). The AMOC is relatively stable during the first 7–10 years, where both ensemble means show differences that are consistent with internal variability. After that, it starts weakening over time in response to the additional meltwater, with the maximum weakening happening during the last 5 years.

The AMOC in density space shows a clearer and much earlier response to the freshwater forcing. The maximum change happens at much higher latitudes, 60.2° N, where the influence of Labrador Sea Waters (LSW) is expected to be stronger. To contextualize the different timings of the responses depending on the vertical coordinate, we note that the computation of the overturning streamfunction in depth space hides the LSW signal on the overturning streamfunction, which does not happen when computing it in density space (Foukal and Chafik, 2024). The ensemble-mean difference in the maximum AMOC at this latitude is similar at the end of the period when compared to the depth space (Fig. 2). However, the signal in the hosing ensemble deviates from the control from the beginning, and its ratio with respect to the internal variability is much higher than in depth space. We can indeed expect the signal to develop faster in density space, as the maximum happens at higher latitudes, closer to the place where the freshwater is injected. The AMOC in density space also shows a more robust response because it shows weaker oscillations due to internal variability, as evidenced in the corresponding ensemble spread.





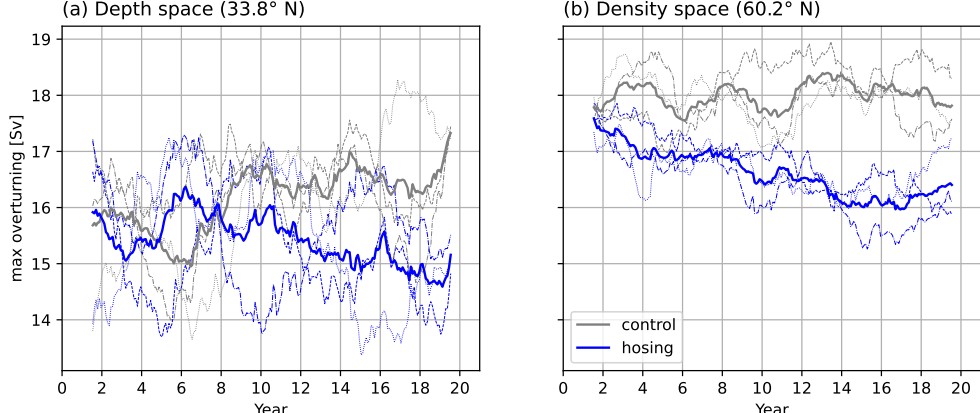

**Figure 2.** Monthly maximum volume overturning streamfunction in (a) depth space at 33.8° N and (b) potential density space at 60.2° N in the Atlantic basin. The solid lines are the ensemble averages, each member and its corresponding control year are plotted with a unique linestyle. The data is filtered with a 36-month (3-year) moving average. The latitudes were selected where the corresponding maximum change in the last 10 years happens, see Fig. 3.

The delayed response in depth space indicates that the freshwater injection needs time to significantly impact the lower latitudes. From now on, we will focus most of our analysis on the last 10 years of the hosing simulation, that is when the AMOC response has fully developed, as shown in Figure 3.

The difference in volume overturning streamfunction in the depth space for the last 10 years between the control and freshwater hosing experiments shows a significant weakening in the AMOC across all latitudes (Fig. 3a). The hosing signal is strongest at around 1000 m and 33.8° N, where the AMOC weakens by 1.4 Sv. The latitudinal band from 40-50° N exhibits an important reduction that exceeds 1 Sv and is the region with the highest difference relative to its mean reference value. By contrast, the lowest latitudes (10-20° N) exhibit a more moderate weakening in the order of 0.5 Sv. Overall, the average reduction between 500-1000 m is about 0.8 Sv, representing a 7 %. This shows that 21 years of hosing is sufficient to produce a detectable weakening in the AMOC, although this is still far from representing a full AMOC collapse.

The AMOC in the density space shows again a stronger and more coherent response across latitudes than in the depth space (Figure 3b). Indeed, the entire 35-62° N band experiences a reduction above 2 Sv, almost twice as big as for the AMOC in depth space. The maximum reduction is located at 60.2° N, where it reaches almost 3 Sv, nearly 20 % of the control climatological benchmark. Overall, the decrease between 1027.4-1027.6 kgm$^{-3}$ is about 10 %. The maximum impact consistently happens from 65 to 10° N for the same density level. This is expected, given that water flows along isopycnals and not isobaths (Foukal and Chafik, 2024).

The impact of the hosing in LSW suggests a reduction of the AMOC through a possible reduction in deep convection in the main North Atlantic deep convection areas: Labrador, Irminger, and Nordic seas. In the next Section, we investigate buoyancy changes in these areas to understand their potential impact on the AMOC. For that, we examine how different thermodynamic variables have changed in the Subpolar North Atlantic (SPNA) in the last 10 years of the simulations.



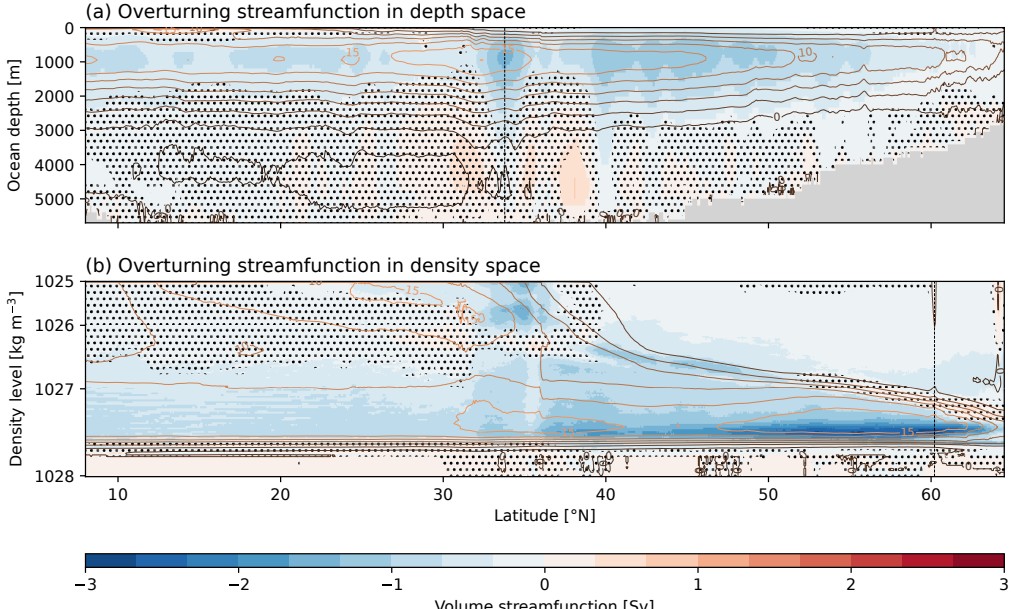

**Figure 3.** Response to the hosing in the last 10 simulated years (i.e. years 12 to 21 since the hosing is started) for the overturning streamfunction in (a) depth space and (b) density space; filled coloured contours represent the ensemble-mean anomalies of the hosing with respect to the reference control and the contour lines the climatology of the control. Non-significant values as identified by the bootstrap methodology are masked with dots to improve the visibility over the significant areas. Vertical dashed lines mark the latitude where the maximum difference happens; $33.78°$ N in depth space and $60.21°$ N in density space.

## 3.2 Large-scale changes in the subpolar region

As expected, the most direct impact of the meltwater forcing on the SPNA is in the salinity field. Figure 4a shows the sea

surface salinity (SSS) anomalies in the last 10 years, which exhibit significant negative values over the whole SPNA area. The largest changes due to the injected freshwater – with SSS losses exceeding 0.4 psu – occur around the Greenland coast and continue along the main boundary currents. Some freshening anomalies are found to penetrate into the Labrador Sea interior, where the average reduction in SSS is close to 0.1 psu. The differences in intensity and significance between the anomalies in the Nordic-Irminger seas and Labrador Sea suggest that the boundary-interior exchange is more important in the Labrador

Sea, which is consistent with previous studies showing that the boundary interior exchanges are more important in the West Greenland Current due to local effects of meanders and eddies (Georgiou et al., 2020; Pacini and Pickart, 2022; Swingedouw et al., 2022; Duyck and De Jong, 2023; Spall et al., 2024). This also suggests that, if a change in surface buoyancy is responsible for the slowdown in the AMOC, this is probably happening in the Labrador Sea. Some of the freshwater anomalies coming out from the Labrador Current appear to reach the North Atlantic Current (NAC), where surface water salinity decreases by more

than 0.1 psu. Significant surface freshening anomalies extend southward to the position of the Gulf Stream in the west (approx.





40° N) and to 35° N in the center and east Atlantic. There is a small imprint in lower latitudes of positive salinity anomalies, likely related to the slowdown in the AMOC and the associated reduction in the northward salinity transport (Fig. B4a).

Sea surface temperatures (SST) also decrease over most of the SPNA and Norwegian Sea, especially along the sea ice edge, the western boundary currents, and the central SPNA (Fig. 4b). Unlike for SSS, no significant temperature changes are found
in the northernmost latitudes of the Greenland coast, discarding any local impact of the meltwater injection. By contrast, the southern Greenland coast and the closest sectors of the Labrador Sea do show significant cooling of up to 2° C. Therefore, this confirms again the importance of the boundary-interior exchange in the Labrador Sea. As for SSS, we see a significant impact on the SSTs in the NAC, which cools down by about 1° C. This cooling is unlikely to be driven by the waters carried out by the Labrador Current, which show a weaker SST response than in the NAC region, and might indeed be explained by the impact
of the already developed AMOC slowdown on the northward heat transport.

The boundary currents around Greenland and the Labrador Sea experience a significant increase in speed (Fig. 4e). At the same time, sea ice concentration grows along the sea ice edge (Fig. 4d), which may be related to changes in northward heat transport (Ma et al., 2024), in the export from the Baffin Bay (Kwok, 2007; Våge et al., 2009; Schiller-Weiss et al., 2024) or persistent surface cooling due to strengthened stratification (Oltmanns et al., 2024).

The surface freshening in the interior Labrador Sea is accompanied by a statistically significant local shoaling of 200-400 m in the mixed layer depth (Fig. 4c). This is about a third of the local climatological mixed layer depth, which implies that the impact is relatively high. The only other significant changes in the mixed layer depth occur along the boundary currents in the northeast and northwest of Greenland, where the mixing also weakens, although the anomalies are smaller. Interestingly, we note a lack of response in the Irminger and Nordic seas, the two other deep convection areas in the NH.

The shoaling in the mixed layer depth does not induce a clear deceleration in the Subpolar Gyre (SPG) strength (Fig. 4f), a connection that has been identified in previous studies (Moreno-Chamarro et al., 2017; Ghosh et al., 2023). However, the barotropic streamfunction shows significant changes over the Gulf Stream (Fig. 4f), characterised by a dipole with positive anomalies North of Cape Hatteras and negative anomalies South of Cape Hatteras. This pattern does not seem to be explained by a change in the position of the Gulf Stream, as diagnostics based on the maximum gradient of surface temperature and surface height show no significant changes in the mean latitudinal position of the current (not shown). Instead, the dipole is
consistent with a slowdown in the Gulf Stream (Fig. 4e) and with the patterns of change in surface temperature and salinity at the NAC (Fig. 4a-b). There is no significant change over the wind stress over the Gulf Stream, which discards any driving role of the atmosphere on the Gulf Stream response (Fig. B3), which may only be linked to a reduction in the deepwater mixing in the SPNA.

To investigate the impact of the freshwater hosing on the ocean subsurface we now focus our analysis on the two main sections where the water mass transport is being measured in the SPNA region, thanks to the collaborative international initiative OSNAP (i.e. Overturning in the Subpolar North Atlantic Program; Lozier et al., 2017). The OSNAP sections (red dashed lines in Fig. 4) cross several regions and currents of interest for our study: The Labrador Current, the Labrador Sea, the Greenland boundary currents, the Irminger Sea, the Reykjanes Ridge, and the Iceland Basin. We particularly focus on the changes in
mixed layer depth, practical salinity, potential temperature, and potential density across the SPNA (Fig. 5). For the OSNAP





**Figure 4.** Response to the hosing in the last 10 simulated years for (a) annual surface salinity, (b) annual surface temperature, (c) March mixed layer depth, (d) DJF sea-ice concentration, (e) annual surface horizontal speed modulus, and (f) annual barotropic streamfunction in the North Atlantic; filled coloured contours represent the ensemble-mean anomalies of the hosing with respect to the reference control and the contour lines the climatology of the control. The contours in (f) have been selected to more clearly identify the positive (purples), zero (black), and negative (greens) levels of the climatological barotropic streamfunction. Non-significant values as identified by the bootstrap methodology are masked with dots to improve the visibility over the significant areas. Red dashed lines mark the OSNAP section. Figure B2 shows an enlarged view of the SPNA.



section, the reduction in mixed layer depth is only significant in the Greenland and Labrador currents and parts of the interior of the Labrador basin (Fig. 5a).

The meltwater fluxes induce statistically significant fresher conditions above 2000 m for almost the whole OSNAP area (Fig. 5b). As expected, the water is much fresher over the continental shelves of Greenland and Newfoundland, where the
boundary currents are located, but there is also a clear penetration of freshwater anomalies into the Labrador Sea interior. In the top 200 m, practical salinity is reduced by about 0.075 psu, followed by a reduction of about 0.05 psu down to 500 m, and a slight significant reduction down to 2000 m. The Deep Western Boundary Current (DWBC), which is located in the western boundary of the Labrador basin at around 500 m depth, does not show a significant change in practical salinity. The intensity of the freshening in the eastern side of the SPNA, east of the Reykjanes Ridge, is similar to that in the Labrador Sea but a bit
lower. The structure of the freshening anomalies in this area follows the one of the isopycnals (see contours in Fig. 5d), which may indicate that the diapycnal mixing in the area is not so effective compared to the effect of the mean flow transport.

The temperature response is more localized than for salinity. There is significant cooling over the Greenland shelf, by more than $1°$ C, and also but not so strong in the Labrador Current (Fig. 5c). Potential temperatures in the upper 200 m of the Labrador Sea interior and the eastern side of the SPNA are about $0.2°$ C colder in the hosing experiments. However, the most
interesting result regarding potential temperature changes happens in the DWBC where there is a significant warming of about $0.3°$ C while no significant changes in salinity occur. This may be linked to the reduction of deep water mixing (Fig. 5a) which feeds the DWBC (Martin-Martinez et al., 2025).

There is a general reduction of density in the upper 2000 m (Fig. 5d). Once again, the Greenland and Labrador currents show the greater changes, with the density gradient increasing towards the interior and accelerating the currents. This is consistent
with previously published results (Schiller-Weiss et al., 2024). The density loss in the boundary currents exceeds $0.15 \ \mathrm{kgm^{-3}}$; whereas in the surface waters of the Labrador Sea interior and over the Reykjanes Ridge, the density anomaly is smaller than $0.05 \ \mathrm{kgm^{-3}}$. These density anomalies are driven by salinity changes, as expected. In other parts of the OSNAP section (e.g. the surface waters of the Irminger Sea and the Iceland Basin), temperature-driven changes compensate for those of salinity, resulting in no clear density response. In contrast, there is no significant salinity response in the DWBC, but rather a positive
temperature anomaly, which drives the density anomaly there.

### 3.3 Time evolution

We now study the time evolution of some key indices since the beginning of the experiments, to understand how the different changes described in the previous Section develop. Figure 6 shows the regions we have used to define the different indices. These regions have been selected taking into account the topography, the position of key ocean currents, and main deep mixing
areas. Since, for a given time step, it will be difficult to measure significance based solely on three data points (i.e. the ensemble members), we will pay particular attention to those changes for which the three members are consistent in sign for at least three consecutive years.

The Greenland Current exhibits a rapid response in surface salinity and temperature (Fig. 7a,b). The salinity reduction is of about 0.4 psu, while the temperature reduction is of around $1°$ C. Among the selected regions, the Greenland Current presents





**Figure 5.** Response to the hosing in the last 10 simulated years for the (a) mixed layer depth, (b) potential temperature, (c) practical salinity, and (d) potential density across the OSNAP section (see red lines in Fig. 4); in (a) anomalies are filled (green) when they are significant by the bootstrap methodology; in (b-d) filled coloured contours represent the ensemble-mean anomalies of the hosing with respect to the reference control and the contour lines the climatology of the control. Non-significant values as identified by the bootstrap methodology are masked with dots to improve the visibility over the significant areas.



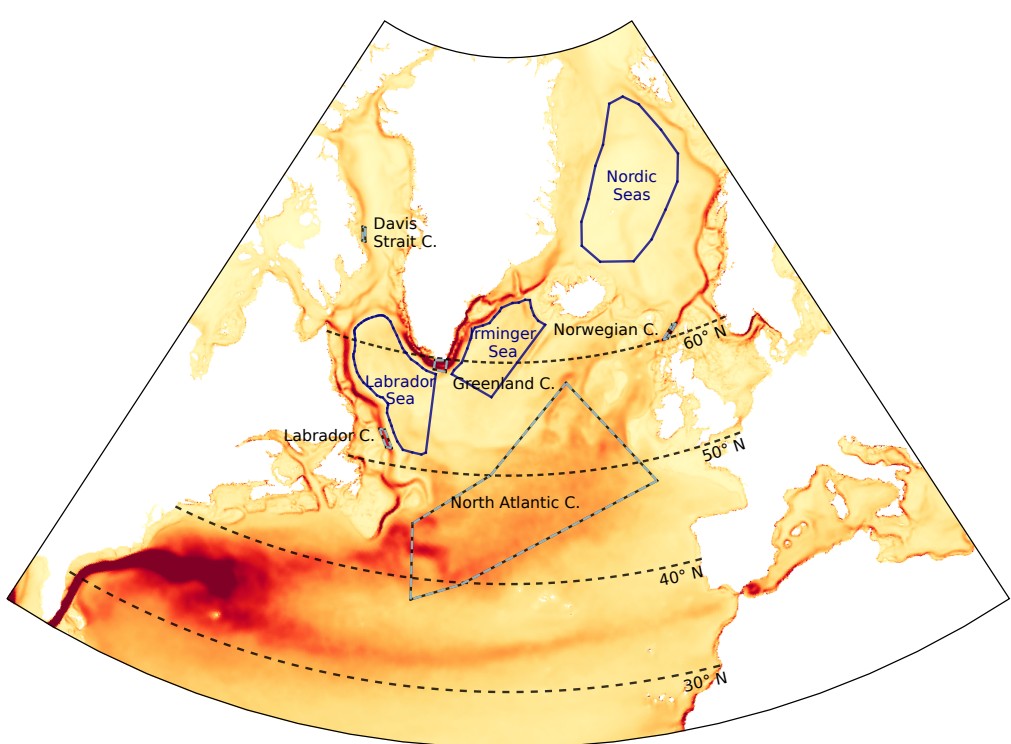

**Figure 6.** Areas used for the time series in Fig. 7. The background represents the average of horizontal speed modulus during the last 10 years in the hosing experiments.

the most immediate response, as it is directly exposed to the meltwater forcing. Farther downstream, the Labrador Current (Fig. 7a) shows consistent changes with the Greenland Current since the beginning of the experiments, although the anomalies are only consistent across members after the first year. Nevertheless, these changes are significantly less intense than for the Greenland Current, with a salinity loss of around 0.15 psu and a temperature drop of approximately 0.2° C. This suggests that the negative Greenland Current salinity anomalies experience some mixing with the neighboring water masses as they move around the Labrador Sea. At the same time, brine rejection due to sea ice growth can contribute to making the current more saline as it advances. It is evident that both currents have also accelerated since the beginning of the freshwater hosing (Fig. 7c).

The Davis Strait Current, located in the Baffin Bay northwest of the Labrador Sea, shows much weaker changes, with little consistency and persistence in time both for the speed and the temperature changes. At the same time, its salinity decreases progressively, with sign-consistent anomalies over the last 5 years of the simulations. Therefore, the changes in this current may respond to the accumulated freshwater in the Baffin Bay coming from the neighboring Greenland coast. At the opposite end of the Arctic-Atlantic sector, the Norwegian Current – which represents an inflow into the Arctic – experiences a consistent





freshening after year 8. The long lag for these salinity changes to emerge compared to the other currents could be explained by the time required by the mean circulation in the SPNA to carry the freshwater anomalies formed near Greenland to that region, although they could also be explained by a delayed SPG weakening and its effects on the northward transport of salt. This second hypothesis, however, is not supported by the barotropic streamfunction, which does not show a clear large-scale reduction in the SPG (Fig. 4f).

A consistent freshening and cooling of the NAC region also takes place after year 11. This coincides with the time at which the overturning circulation shows a consistent drop of more than 1 Sv at 40°, 50°, and 60° N (Fig. 7e). Therefore, the changes in the NAC region may just emerge in response to the reduction in the northward transport of warm and saline waters due to the large-scale AMOC slowdown. This particular response may trigger a feedback that further reduced salinity in the whole SPNA, as it has already been described in several studies (Drijfhout et al., 2011; Jackson, 2013; Liu et al., 2014). The horizontal speed in the NAC region shows a consistent reduction between years 10 and 15, and, although the mean anomaly remains negative thereafter, the ensemble members no longer agree on the sign anomaly.

To understand the changes in the overturning streamfunction we now focus on the mixed layer depth changes. Only the Labrador Sea mixed layer depth gets consistently reduced over time, in contrast to the Irminger and Nordic Seas (Fig. 7d) where no consistent changes are observed. Both the Irminger and Nordic seas have shallow mixed layer depths in the control experiment, the Labrador Sea being the main site of deep convection in the SPNA in EC-Earth3P-VHR simulations (Martin-Martinez et al., 2025). In the Labrador Sea, the reduction becomes consistent in sign from year 7 onwards, with an average reduction of 100 m. The changes in the mixed layer depth seem to lead by 2 to 4 years the changes in the maximum overturning streamfunction at 40° and 50° N, but lag by 2 years the AMOC changes at 60° N, which might be explained instead by the thermal wind response to the salinity-driven density changes along the Greenland and Labrador currents. The signal does not reach 30° N with the same intensity and persistence, and it is only consistent for years 12-16.

To better explore the changes in the Labrador Sea, we show the Hovmöller diagrams of the spatial averages of practical salinity, potential temperature, and potential density (Fig. 8) in the Labrador Sea interior as a function of time and depth. For the three variables, changes are stronger in the upper 200 m than below. The salinity changes are not fully consistent in time nor across the ensemble members during the initial 7 years. In fact, during these years, there is a competition between the salinity and temperature anomalies — which are consistently negative throughout the whole period — to drive the changes in density, which results in a succession of positive (when temperature dominates) and negative (when salinity dominates) changes at the surface. However, from the year 7 onwards, salinity changes completely dominate the changes in density in the upper 500 m of the Labrador Sea interior. This happens at the same time that consistently negative anomalies in the mixed layer depth start developing (Fig. 7d). We hypothesize that the faster response of surface temperature in the Labrador Sea interior arises because lateral advection of heat (and thus temperature anomalies) is more efficient than the advection of salt/freshwater anomalies. The latter are more prone to dilution and mixing along the boundary, requiring several years of cumulative transport before significantly impacting interior salinity.

The mixed layer depth response can be explained by a change in the stratification of the Labrador Sea, as the freshwater accumulates mainly at the surface, increasing the vertical salinity gradient, which leads during the last years to a more stratified





**Figure 7.** Time series of ensemble-mean anomalies due to the hosing with respect to the reference control for (a) annual surface salinity, (b) annual surface temperature, (c) annual surface horizontal speed modulus, (d) march mixed layer depth, and (e) annual maximum overturning in density space. The areas used in (a-d) are shown in Fig. 6. All the data is filtered with a 3-years moving average. Dots denote the points where the difference of the three members agree in the sign.





density distribution (Fig, 8d). The density changes induced by the potential temperature vertical profile oppose those induced by salinity (Fig, 8e), without fully counterbalancing them (Fig, 8f). These results support a weakening response of Labrador
Sea interior deep water mixing to the quasi-realistic Greenland meltwater forcing mediated via freshwater exchanges from the boundary currents. Interestingly, Wei and Zhang (2024) was able to produce the contrary effect when injecting only freshwater in the south of the Nordic Seas. Similarly, Ma et al. (2024) demonstrated the varying impact of four freshwater injection areas in the North Atlantic, with freshwater injected in the Irminger Basin being the most effective at reducing deep convection in the Labrador Sea. Therefore, the distribution of the forcing may play an essential role, whereas other idealised experiments,
such as the uniform forcing from Jackson et al. (2023), may not produce a realistic response in the SPNA dynamics.

## 3.4   Global impacts

Previous studies have highlighted important global impacts in different hosing experiments, generally related to a collapse or substantial weakening of the AMOC causing a widespread, intense cooling of the North Atlantic (Jackson et al., 2015; Liu et al., 2020; Orihuela-Pinto et al., 2022; Bellomo et al., 2023; van Westen et al., 2024). These studies have also identified
changes in remote regions, such as the ITCZ (Jackson et al., 2015; Liu et al., 2020; Orihuela-Pinto et al., 2022; Bellomo et al., 2023; Ma et al., 2024) and Antarctic temperatures (Liu et al., 2020; Orihuela-Pinto et al., 2022; Ma et al., 2024; Diamond et al., 2025). We extend our analysis to the global impacts in our hosing simulations, acknowledging that the limited length of our simulations (21 years) and the moderate but consistent AMOC weakening might not be sufficient to drive large-scale, strong impacts on the global climate. To study these global impacts, we examine changes in the atmospheric surface temperature, sea
level pressure, and precipitation. Figure 9 shows the freshwater hosing-driven anomalies in these three variables over the last ten years for the months of December-January-February (DJF, boreal winter or austral summer) and June-July-August (JJA, austral winter or boreal summer).

The most intense surface air temperature changes occur over the ocean during boreal winter. During this season local surface air temperatures drop by more than 5° C in the northern Labrador Sea and by about 2.5° C in the Nordic Seas. There is also
a slight consistent cooling in most of the SPNA and Greenland. These negative changes in atmospheric surface temperature are linked to similar changes in SST (Fig. 4b). Over the continents, the most clear signal is a widespread cooling from the Eastern Mediterranean to the Middle East. During the austral winter the strongest response happens in the SH, with a strong warming (more than 5.5° C) over the Amundsen and Ross Seas, near West Antarctica, which also occurs but weaker in the austral summer . Similar positive temperature anomalies in the Amundsen Sea are described by Diamond et al. (2025) in an
analysis focused on NAHosMIP runs. The main signals in the boreal summer in the NH are a widespread cooling in the SPNA (of smaller amplitude than for winter) and a warming in the Western Mediterranean. Overall, the described anomalies in the NH and SH are consistent in sign during summer and winter. However, the summer anomalies tend to be weaker in magnitude, usually less than half the intensity of the winter ones, in particular in the deep convection areas of the Labrador and Ross seas, where the winter mixing response might reinforce the temperature signals.

The sea level pressure response is most prominent in the polar regions. During DJF, the polar low gets reinforced, with a local minimum also observed over the Labrador Sea accompanied by two anomalous high pressure systems over Central North



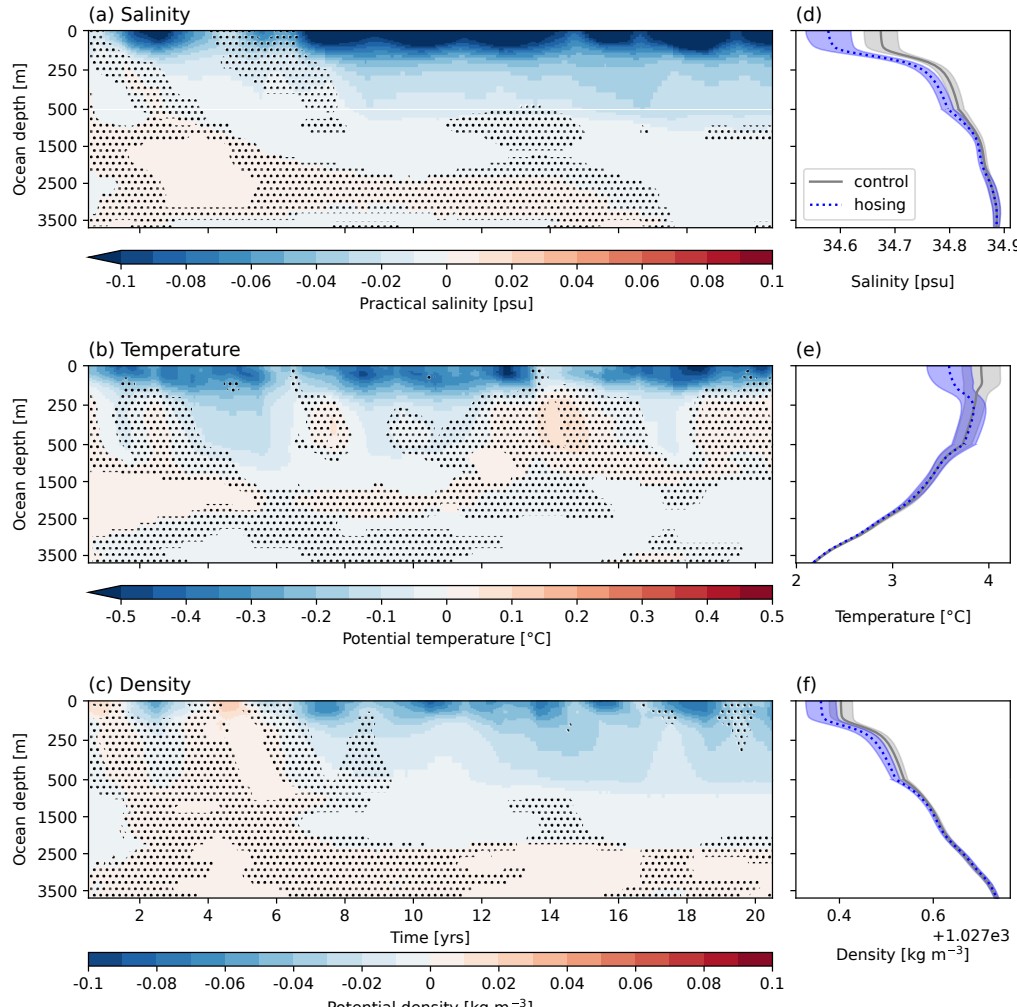

**Figure 8.** Depth vs time Labrador Sea interior ensemble-mean anomalies in (a) salinity, (b) temperature and (c) density as computed between the hosing and the reference control. The anomalies are computed as 12-month moving averages. Anomalies that are not fully consistent in sign between the three ensemble members are masked with dots to highlight the visibility of the fully consistent ones. Vertical profiles of the spatially averaged DJF Labrador Sea interior (d) salinity, (e) temperature and (f) density values in the hosing (blue) and control (grey) ensembles during the last 10 years of the simulations. The ensemble means are shown by lines and the spread represents one standard deviation (in the member space) around that mean.





America and Europe. The anomalous high over Europe is consistent with a deflection of the westerlies that could be driving northerly cold-air advection into the Eastern Mediterranean, contributing to the observed cooling there. We also see positive high pressure anomalies in most of the Antarctic Circle, opposed by negative anomalies in the Pacific sector of the Southern Ocean. These changes may be associated with a positive Northern Annular Mode (NAM) and a negative Southern Annular Mode (SAM) response. The NAM response could be reinforcing the cooling over the Labrador Sea, as it is consistent with an intensification of the westerlies over that region, while counterbalancing effect might be happening between SAM and the cooling signal over the Southern Ocean, as negative SAM phases tend to weaken the westerlies and their forcing on the ocean surface.

During JJA, the responses in sea level pressure are weaker than for DJF in the NH. There is a consistent increase in sea level pressure over the central North Atlantic region and a decrease over Greenland, which jointly support an anomalous positive phase in the summer North Atlantic Oscillation (NAO). Positive responses are also observed over the Urals and West Siberia. In the SH, there is a big negative response in the Southern Pacific and Southern Atlantic regions and large positive responses south of Australia and over the Amundsen Sea. This last change could explain the positive temperature anomaly previously seen over the same region and season, as the warming is located to the west of the sea level pressure anomaly, where warm southward advection is expected. These changes could be induced by a wave train, as shown by Diamond et al. (2025).

Regarding precipitation in DJF, we see a consistent increase south of the Equator, both in the Pacific and Atlantic basins (Fig. 9e,f), which could be indicative of a southward migration of the ITCZ. However, the precipitation signal is more regionally confined than show in other models in response to stronger AMOC weakenings (Jackson et al., 2015; Liu et al., 2020; Orihuela-Pinto et al., 2022; Bellomo et al., 2023; Ma et al., 2024). Also, a more detailed analysis with diagnostics that characterises its latitudinal shifts and intensity at the global scale (as in Santos-Espeso et al., 2025) does not reveal a consistent ITCZ response (not shown). We also notice a precipitation increase in the equatorial Atlantic during JJA and a decrease in the Amazonas basin and the Maritime continent in DJF.

## 4 Conclusions

This paper explores the impact of applying a quasi-realistic Greenland meltwater forcing in the Subpolar North Atlantic (SPNA) region, the Atlantic Meridional Overturning Circulation (AMOC), and the global climate. To that end, we use the HighResMIP coupled global model EC-Earth3P-VHR, which has an eddy-rich ocean grid (about 8 km in the mid-latitudes), to run a three-member ensemble of 21 year long freshwater hosing experiments, branching from different AMOC states of a 1950-control simulation. The magnitude of the forcing corresponds to 0.04 Sv of freshwater on an annual average, with a temporal and spatial distribution around Greenland derived from observations.

We find the following responses to the Greenland freshwater hosing:

- A reduction of up to 20 % of the AMOC transport by the end of the experiments, corresponding to 2-3 Sv between 35° N and 62° N when measured in density space. This is far from representing a collapse but is a substantial reduction given the relatively short time span and moderate strength of the forcing applied.





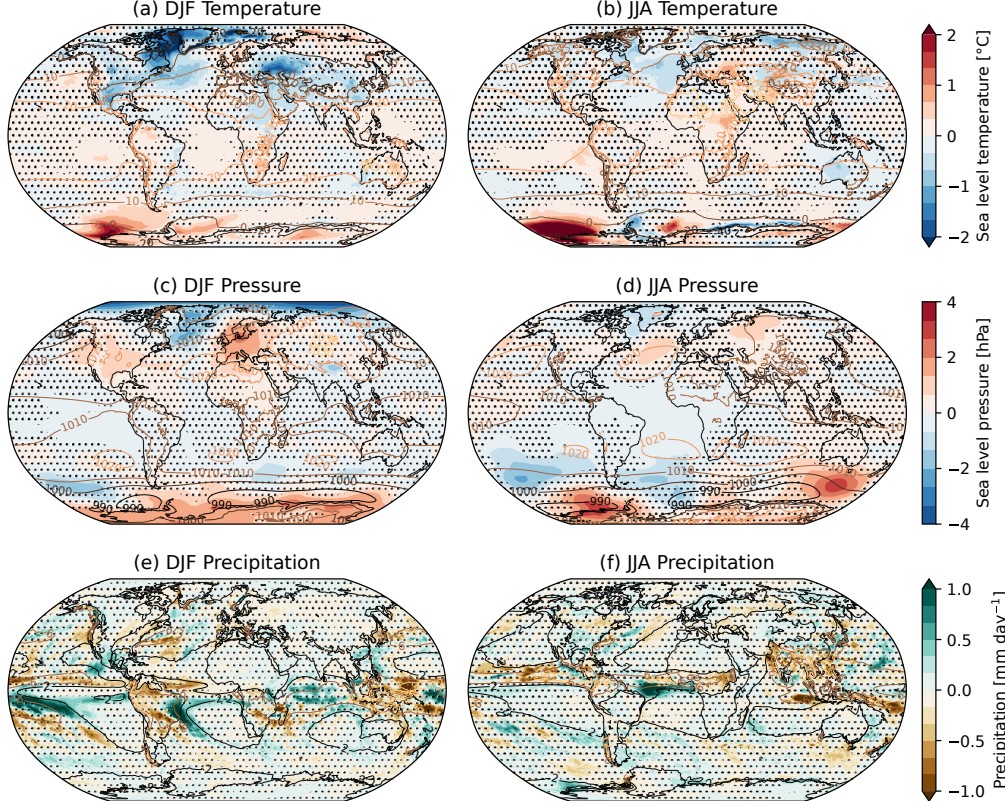

**Figure 9.** Response to the hosing in the last 10 simulated years for (a-b) atmospheric surface temperature, (c-d) sea level pressure, and (e-f) precipitation, for (a-e) DJF and (b-f) JJA; filled coloured contours represent the ensemble-mean anomalies of the hosing ensemble with respect to the reference control and the contour lines the climatology of the control. Non-significant values as identified by the bootstrap methodology are masked with dots to improve the visibility over the significant areas.

- A general surface freshening and cooling spanning across the SPNA. While the changes along the boundary currents and the interior Labrador Sea seem to directly emerge in response to the freshwater forcing, the changes in the eastern SPNA seem to be linked to a reduction in the northward transport of salty and warm waters, driven by the slowdown of the AMOC.

- In the first year, the freshwater fluxes induce an acceleration of the boundary currents, which drives the local cooling. Over the following years, freshwater anomalies slowly penetrate into the Labrador Sea interior where they start reducing the vertical mixing by year 7. The AMOC experiences a first rapid weakening in the SPNA - via thermal wind balance - to the signals emerging along the boundary current, a weakening that is enhanced and becomes more widespread as soon as the Labrador Sea mixing responds.



    – At the global scale, only few regions experience significant impacts, like a positive phase of the Northern Annular mode and a negative phase of the Southern Annular mode in the boreal winter, which cause a cooling over Western Europe and a warming over the Amundsen and Ross Seas. Typical impacts previously linked to strong AMOC reductions in freshwater hosing experiments, like a southward shift of the Intertropical Convergence Zone (ITCZ), are not reproduced in our relatively short EC-Earth3P-VHR experiments.

The fidelity of the spatio-temporal distribution applied for the injected meltwater fluxes and the model capacity to resolve mesoscale ocean processes are two key aspects of our study, both jointly expected to enhance the realism of the simulated local and global impacts of the Greenland freshwater hosing. The narrow boundary currents around Greenland quickly carry the freshwaters southward, while mesoscale eddies contribute to the lateral exchanges with the interior of the Labrador Sea (Georgiou et al., 2020; Schiller-Weiss et al., 2024) while keeping the interior of the Irminger Sea isolated. Meanwhile, freshwater anomalies are carried south through the Labrador Current until it meets the Gulf Stream. There, an accurate location of the Gulf Stream separation, which is known to critically improve at eddy-rich resolutions (Marzocchi et al., 2015; Moreno-Chamarro et al., 2021; Frigola et al., 2025), is essential for accurately modelling the interactions with the Gulfstream and its subsequent impact on the northward heat and salinity transport.

Interestingly, the local response detected in the SPNA in our experiments aligns with that found by Schiller-Weiss et al. (2024), who used a global configuration of the ocean/sea-ice NEMO model with a finer horizontal resolution of $1/20°$ in the North Atlantic, where they studied the response of the SPNA to the 1997-2021 enhanced Greenland melting. We further highlight that the identified impacts are consistent across the three members, suggesting that the initial state of the AMOC does not critically determine the main ocean response. The atmospheric responses tend to show larger differences across members, which suggests that longer experiments might be needed for them to emerge more robustly, like the typical southward shift of the ITCZ (Jackson et al., 2015; Liu et al., 2020; Orihuela-Pinto et al., 2022; Bellomo et al., 2023; Ma et al., 2024), which might require a stronger AMOC response. Likewise, a new coordinated protocol to assess the sensitivity of the AMOC to the projected Greenland ice sheet melting tailored to eddy-rich climate model configurations is warranted to determine the inter-model consensus and main underlying uncertainties in the AMOC response, which is key to better understand its future evolution. The Tipping Points Modelling Intercomparison Project (TipMIP; Winkelmann et al., 2025) could define and coordinate this protocol, building on the experimental frameworks put forward by the High-Resolution Model Intercomparison Project phase 2 (HighResMIP2; Roberts et al., 2025).

*Code and data availability.* The control experiment data (EC-Earth Consortium (EC-Earth), 2024) are available from the Earth System Grid Federation (ESGF, https://esg-dn1.nsc.liu.se/search/cmip6-liu/; last access: 26 November 2025). The hosing experiment last 10 annual means (years 12–21) of some 2D variables and last 10 years climatology of some 3D variables data are available at EERIE's Zenodo (Ortega et al., 2025). Monthly data of the hosing experiment could be shared through an FTP upon request.

Used ESMValTool recipes and diagnostics will be made available in the revised version.





**Appendix A: Estimation of the temperature**

Due to a failure in the data storage, many global 3D temperature data files were lost. Therefore, Fig. B4 shows an estimate of the temperature instead of the model output. The estimation was performed using an approximation based on constant coefficients of thermal expansion ($\alpha$) and salinity contraction ($\beta$). Given the relation in equation (A1):

$$d\rho = \rho_0 \left( \beta \, dS - \alpha \, dT \right) \tag{A1}$$

we can linearly approximate the temperature as shown in equation (A2):

$$T = T_0 + \frac{1}{\alpha} \left( \beta \left( S - S_0 \right) - \frac{\rho - \rho_0}{\rho_0} \right) \tag{A2}$$

we have taken the coefficients from Nycander et al. (2015) suggested by Vallis (2006); $\alpha = 1.67 \cdot 10^{-4}\,\mathrm{K}^{-1}$, $\beta = 7.8 \cdot 10^{-4}\,\mathrm{psu}^{-1}$, $T_0 = 10°\,\mathrm{C}$, $S_0 = 35\,\mathrm{psu}$, $\rho_0 = 1027\,\mathrm{kg\,m}^{-3}$.

Note that the relation in equation (A1) is non-linear, as $\alpha$ is defined at constant salinity and $\beta$ at constant potential temperature. Therefore, the estimated temperature resulting from (A2) is not exact. However, a test in the OSNAP section shows that the intensity of the anomalies is smaller, mainly in the regions where the anomalies are greater. However, the sign and significance of the anomalies remain the same.

**Appendix B: Supplementary figures**



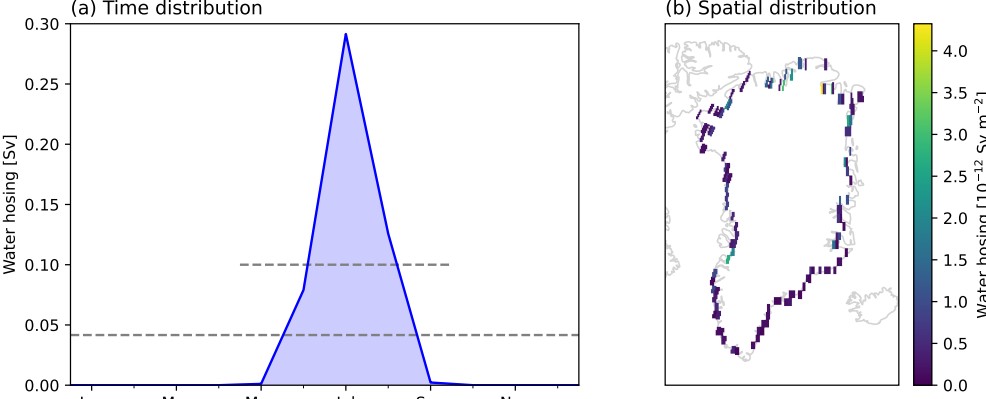

**Figure B1.** (a) Total monthly freshwater forcing time distribution and (b) May-September averaged freshwater forcing runoff spatial distribution. Values given to the model as an input. The dashed lines in (a) indicate the annual average and the average over the five months of forcing.





**Figure B2.** As Fig. 4, showing an enlarged view of the SPNA.





**Figure B3.** Response to the hosing in the last 10 simulated years for (a) surface horizontal speed modulus, (b) barotropic streamfunction, (c) x-ward wind stress, and (d) y-ward wind stress; filled coloured contours represent the ensemble-mean anomalies of the hosing with respect to the reference control and the contour lines the climatology of the control. Non-significant values as identified by the bootstrap methodology are masked with dots to improve the visibility over the significant areas. Red dashed lines mark the OSNAP section.





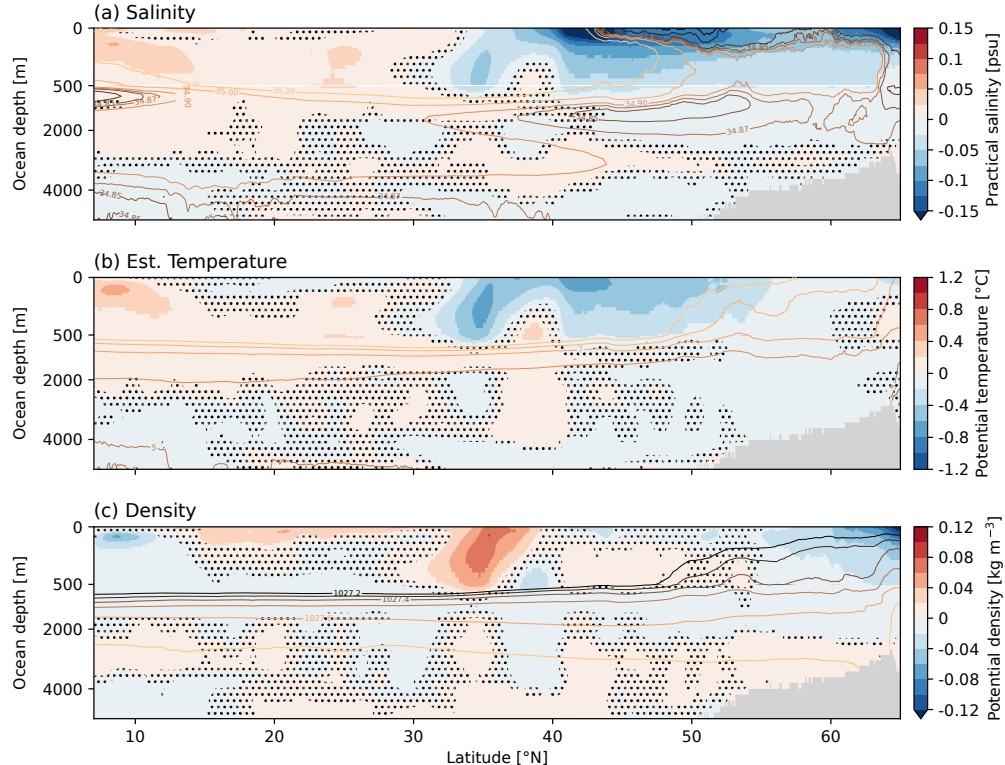

**Figure B4.** Response to the hosing in the last 10 simulated years for (a) salinity, (b) estimated temperature, and (c) density zonally averaged in the Atlantic basin; filled coloured contours represent the ensemble-mean anomalies of the hosing with respect to the reference control and the contour lines the climatology of the control. Non-significant values as identified by the bootstrap methodology are masked with dots to improve the visibility over the significant areas.

410    *Author contributions.* EMM carried out the analysis and wrote the manuscript. EMC, FG, JSVS, CAP, and PO suggested analysis and gave inputs to the manuscript. PAB and DK post-processed and cmorized the model data. SLT gave support using ESMValTool and made improvements for memory usage and performance in the tool needed for the analysis. EMM, EMC, and PO designed the protocol. EMC implemented the freshwater forcing and ran the simulations.

*Competing interests.* The authors declare that they have no conflict of interest.

415    *Disclaimer.* Views and opinions expressed are, however, those of the authors only and do not necessarily reflect those of the European Union or the European Climate Infrastructure and Environment Executive Agency (CINEA). Neither the European Union nor the granting authority can be held responsible for them.



*Acknowledgements.* We would like to thank Amanda Frigola, Alba Santos-Espeso, Bernardo Maraldi, Marta Brotons, and Rein Haarsma for their constructive feedback on the analyses carried out. We also thank Marion Devilliers and Didier Swingedouw for their guidance in defining the experimental protocol. We acknowledge the scientific and technical support from colleagues in the BSC's Climate Variability and Change and Computational Earth Sciences groups. We also value the ESMValTool development team for their work and support. We appreciate the feedback received at conferences and external events.

This publication is part of the EERIE project funded by the European Union (grant agreement no. 101081383).

This work has received funding from the Swiss State Secretariat for Education, Research and Innovation (SERI) (contract no. 22.00366).

This work was funded by UK Research and Innovation (UKRI) under the UK government's Horizon Europe funding guarantee (grant nos. 10057890, 10049639, 10040510, and 10040984).

Eneko Martin-Martinez received funds from grant no. PRE2021-097163 funded by MCIN/AEI/10.13039/501100011033 and by ESF Investing in Your Future.

Eneko Martin-Martinez and Eduardo Moreno-Chamarro received funds from grant no. PID2020-114746GB-I00 funded by MICIU/AMEI/10.13039/501100011033.



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
