# Peer review of "North Atlantic response to a quasi-realistic Greenland meltwater forcing in eddy-rich EC-Earth3P-VHR hosing simulations"

_EGUsphere, 2025_

## Referee Comment (RC1)

**Review of the publication:**

**North Atlantic response to a quasi-realistic Greenland meltwater forcing in eddy-rich EC-Earth3P-VHR hosing simulations**

Eneko Martin-Martinez, Eduardo Moreno-Chamarro, Fraser William Goldsworth, Jin-Song von Storch, Cristina Arumí-Planas, Daria Kuznetsova, Saskia Loosveldt-Tomas, Pierre-Antoine Bretonnière, and Pablo Ortega

**General assessment and recommendation**

This study investigates the response of the Subpolar North Atlantic, the AMOC, and selected global climate indicators to a freshwater perturbation using the eddy-rich global climate model EC-Earth3P-VHR. By combining high spatial resolution with a spatially and temporally distributed freshwater forcing around Greenland, the authors provide a detailed analysis of boundary-current pathways, boundary–interior exchanges, and the time evolution of the AMOC response.

The manuscript is rich, technically sophisticated, well-written and novel in several respects. In particular, the use of an eddy-rich global model allows the authors to resolve narrow boundary currents and mesoscale processes that are poorly represented in most previous hosing experiments. The analysis provides valuable insight into how freshwater anomalies are advected along boundary currents, gradually penetrate the Labrador Sea interior, and ultimately lead to a weakening of the AMOC. The exploration of associated atmospheric responses further broadens the scope of the study.

Overall, the paper addresses an important and timely question concerning the sensitivity of the AMOC to Greenland freshwater input, and the results have the potential to make a meaningful contribution to the literature. However, **several key methodological and interpretational aspects require clarification** before the conclusions can be considered fully robust. In particular, the **effective magnitude and realism of the freshwater forcing**, the distinction between imposed and background freshwater fluxes, and the physical mechanisms underlying the relatively rapid AMOC response need to be more clearly documented and discussed. Addressing these issues is essential to ensure that the simulated AMOC weakening is interpreted correctly and placed in an appropriate physical and observational context. For these reasons, I recommend major revision.

**Major comments**

A central issue that needs to be addressed more explicitly concerns how much freshwater is actually added to the ocean in the experiments, and how this compares to observed Greenland freshwater fluxes.

While the manuscript states that a freshwater flux of 0.04 Sv is applied based on Greenland meltwater estimates, this flux appears to be added on top of the model's background freshwater inputs, including runoff and iceberg (calving) fluxes already present in the control simulation. However, the magnitude and temporal evolution of these background fluxes are not documented. As a result, it is currently unclear what the total freshwater input experienced by the ocean actually is in the hosing experiments.

This point is critical for two reasons:

- without explicitly documenting the control-run runoff and calving fluxes, it is not possible to determine whether the AMOC response corresponds to an additional 0.04 Sv or to a substantially larger total freshwater perturbation. Given that observed Greenland freshwater fluxes around 1950 are closer to ~0.03 Sv or lower (runoff plus solid ice discharge), the applied forcing may significantly exceed realistic values once background fluxes are included.
- all simulations are conducted under constant 1950 forcing, rather than transient historical. Combined with the relatively strong freshwater input, this implies that the experiments should be interpreted as an idealized sensitivity study, rather than as a fully realistic representation of recent or near-future Greenland melt conditions. This distinction should be made explicit throughout the manuscript, including in the abstract and conclusions.

In addition, given the slow advective timescales of deep water masses, it would be helpful to clarify which mechanisms allow the AMOC and associated northward transports to respond within approximately 10 years. In particular, the authors should distinguish more clearly between:
- dynamical circulation adjustments (e.g. thermal wind balance, pressure-field and boundary-current adjustments), and
- the physical propagation of deep water mass anomalies, which occurs on much longer timescales. Clarifying this distinction would significantly strengthen the physical interpretation of the results and avoid potential confusion between transport changes and tracer propagation.

To address these issues, I strongly recommend that the authors **provide time series of Greenland runoff and iceberg (calving) freshwater fluxes in the control ensemble**, clearly state the total freshwater flux applied in the hosing experiments (background + additional forcing), explicitly compare this total flux to observed estimates, and clearly frame the experiments as idealized sensitivity experiments under constant 1950 forcing, rather than fully realistic.

Addressing these points would substantially improve the transparency, physical consistency, and interpretability of the manuscript.

**Minor Comments**

**Abstract**

The abstract states that the model is forced for 21 years, add if this forcing is applied only at the surface or also distributed at depth.

The abstract mentions "an acceleration and a cooling". Please clarify what is accelerating (e.g. boundary currents, gyre circulation) and specify the spatial scale. Similarly, please describe the origin of the cooling (e.g. reduced northward heat transport, enhanced stratification), as the freshwater itself is not necessarily colder than the ambient ocean.

The phrase "along the currents" is vague. Please specify which currents are being referred to (e.g. Greenland boundary currents, Labrador Current, North Atlantic Current).

The statement "lead to a rapid weakening of the AMOC" would benefit from clarification. Surface cooling alone would tend to increase density and potentially strengthen the AMOC; please clarify how the combined effects of freshening, stratification, and cooling result in a net AMOC weakening.

The sentence "By the end of the simulation, the AMOC has weakened by almost 3 Sv at subpolar latitudes (i.e. a decrease of around 20 %), with an average relative decrease of 10 % for the whole Northern Hemisphere" is unclear. Please specify the latitudes at which these values are computed and provide corresponding absolute AMOC values to avoid ambiguity.

For clarity, the abstract should explicitly state the total freshwater flux applied (including background runoff and calving) and how this amount compares to observed Greenland freshwater fluxes, as this directly conditions the magnitude of the AMOC response.

In addition, the abstract should explicitly state that the simulations are conducted under constant 1950 forcing, rather than transient historical, as this strongly conditions the interpretation of the results and the realism of the freshwater forcing.

**Introduction**

General remark: the Introduction tends to make strong causal assumptions, sometimes blurring the distinction between what is directly observed, what is inferred from models, and what remains speculative. A more cautious framing of these mechanisms would improve the scientific clarity and consistency of the manuscript.

l.18: The statement "The associated changes in salinity modify the surface density" would benefit from a citation. It is not clear that this link has been directly observed in the context discussed here, so please support this statement with an appropriate reference.

l.19: The link between freshwater forcing and deep water formation is presented in a very direct way. Consider explicitly mentioning the role of upper-ocean stratification and deep convection processes, so that the physical mechanisms and underlying assumptions are clearly stated.

l.22: The phrase "whether the projected AMOC reduction will result in its complete shutdown" is ambiguous. Please clarify whether this refers to continuous anthropogenic forcing in general or specifically to freshwater forcing.

l.39: The expression "using idealized model experiments" to describe Devilliers et al. (2021, 2024) is misleading. These studies use realistic, spatially and temporally distributed freshwater fluxes derived from observations rather than idealized hosing. Please revise this wording accordingly.

l.60: The sentence "In order to make the most of the finer resolution, a protocol with a greater focus on the imminent transient response than on long-term equilibrium is needed" does not clearly follow from the previous sentence and introduces a new topic. Consider separating this into a new paragraph and reformulating to clarify that the long-term response is difficult to explore at such high resolution due to computational constraints. It would also be useful to briefly discuss the role of internal variability and the need for large ensembles to separate forced signals from internal variability on long timescales, compared to the shorter timescales targeted here.

l.61: The sentence "An overly idealized Greenland hosing configuration can result in an unrealistic distribution of injected freshwater" would benefit from clarification. Please briefly explain what aspects of the hosing configuration are emphasized in Goldsworth (2026) to obtain a more realistic response, and consider also referencing and describing the recent protocol proposed by Schmidt et al. (2025, https://doi.org/10.5194/gmd-18-8333-2025).

l.63: Replace "This" with "This protocol" for clarity.

**2 Methodology**

**2.1 Experimental set-up**

l.83: The statement "based on observations from Bamber et al. (2018)" is not fully accurate. The dataset in Bamber et al. (2018) is derived from a combination of observations and high-resolution regional models. Please correct this wording here and consistently throughout the manuscript.

l.85: The conversion from 0.04 Sv corresponds to approximately 1261 km³ yr⁻¹, not 1322 km³ yr⁻¹. Please correct this value.

l.86: The sentence "close to estimations for the values from 2013–2016 (Bamber et al., 2018)" is ambiguous. Please clarify whether this refers to the total Greenland freshwater flux or only to specific components (e.g. solid ice discharge plus runoff, including ice and tundra contributions).

l.88: The statement "We include the freshwater as an additional term to model river runoff" raises an important point that needs clarification. The manuscript does not specify the magnitude of runoff and iceberg (calving) freshwater fluxes in the control run. If the hosing flux is added on top of these existing fluxes, the experiment evaluates the impact of 0.04 Sv plus the model's background freshwater input, rather than 0.04 Sv alone. Please specify the control-run freshwater fluxes and consider providing a time series of runoff and iceberg discharge for the control experiment.

l.90: The phrase "as well as vertically" is vague. Please describe the vertical distribution of the freshwater input (e.g. depth range, uniform or non-uniform distribution).

l.97: The terms "weak, moderate and strong AMOC states" would benefit from quantitative definitions. Please provide representative values (e.g. in Sv) at standard latitudes such as 26° N and/or 45° N to clearly distinguish these regimes.

l.99: The sentence "To avoid the effects of a strong initial model drift" is unclear. Since the control and hosing simulations use identical forcings, it is not obvious why the control run would drift relative to the spin-up. Please clarify the origin of this drift.

l.101: The statement that "The three hosing runs use the same […] constant 1950-forcing" is key information and should be clearly stated in the abstract. The experiments are conducted under fixed control forcing rather than transient forcing, which is not fully consistent with the term "realistic conditions" currently used. This distinction should be made explicit in the abstract.
In addition, freshwater fluxes around 1950 were substantially lower than 0.04 Sv (approximately 900 km³ yr⁻¹, i.e. < 0.03 Sv, for runoff plus solid ice discharge in Bamber et al., 2018). In the present experiments, 0.04 Sv is applied in addition to the model's background runoff and calving, resulting in a total freshwater input that is significantly larger than realistic 1950 values. This discrepancy should be clearly acknowledged and discussed, as it conditions the interpretation of the results.

**2.2 Results evaluation**

This subsection would benefit from being rewritten to clarify the evaluation methodology and improve overall readability.

l.109: The sentence "The anomalies between the hosing and control experiments are computed by matching the years between both experiments after the initialization" is unclear. Please specify what

is meant by "matching the years" (e.g. identical simulation years after initialization, fixed lag, or alignment relative to the hosing onset), and explicitly state how many years are matched.

l.110: The phrase "to remove any remaining model drift" would benefit from clarification. Please specify the origin of the drift being removed (e.g. background model drift, adjustment to freshwater forcing, or both).

l.110: The expression "align the initial internal climate variability" is unclear. Please clarify what is being aligned (e.g. phase of internal variability, ensemble mean state) and how this alignment is performed.

l.115: The reference to "the previous sample of anomalies" is ambiguous. Please clarify which sample is being referred to and how it is defined.

l.120: The choice of a three-year threshold ("for at least three years") appears arbitrary unless motivated by previous studies. If this choice is methodological, please state it explicitly, for example by rephrasing to "we assume here that three years is sufficient to…" or "we propose to use a three-year threshold to…".

**3. Results**

General remark: the term 'volume overturning streamfunction' is technically correct but somewhat redundant, as overturning streamfunctions are by definition volumetric. Consider simplifying to 'overturning streamfunction' for clarity and consistency with standard usage.

**3.1 Meltwater impact on the AMOC**

l. 124: "the time evolution of the maximum overturning" on annual means? The caption says "Monthly maximum" which is unclear.

l. 122–124 + Fig. 2 caption: The latitudes 33.8° N (depth space) and 60.2° N (density space) are selected based on where the maximum change occurs during the last 10 years. Since these latitudes are chosen a posteriori, it would be helpful to clarify that they are intended as diagnostic locations rather than representative of the basin-wide AMOC response, or to briefly discuss the sensitivity of the results to this choice.

l.123–127: the AMOC is described as "relatively stable" during the first 7-10 years, while differences between ensembles are said to be consistent with internal variability. The term "stable" may be ambiguous here. A formulation explicitly referring to the absence of a statistically significant signal relative to internal variability might improve clarity.

l.128: it is stated that the AMOC response in density space is "clearer and much earlier" than in depth space. While this is convincing qualitatively, the argument would be strengthened by a more quantitative criterion (e.g., emergence time or a significance threshold).

l.133: the method used to quantify this ratio of the signal to internal variability is not specified. Please clarify how internal variability is estimated (e.g., ensemble spread or temporal variance) and how this ratio is computed.

l.128–138 and Fig. 2: comparison between depth space and density space responses involves different latitudes (33.8° N versus 60.2° N). As a result, the contrast reflects both differences in

vertical coordinate and latitude. A brief clarification disentangling these two effects, or explicitly stating this limitation, would strengthen the interpretation.

l.150: the statement that the stronger coherence in density space is expected because "water flows along isopycnals and not isobaths" is broadly correct, but somewhat simplified since diapycnal transformations are also central to the AMOC, a more nuanced phrasing is needed.

l. 146: when stating that 21 years of hosing produces a detectable AMOC weakening, it would be useful to briefly contextualize the magnitude of this response relative to the imposed freshwater flux (add the amount of the total freshwater fluxes imposed to the model) and to previous freshwater hosing studies (Swingedouw et al 2022, Jackson et al 2023, Van Westen 2024).

l. 154: regarding the link between LSW, deep convection, and AMOC weakening, it is plausible but remains qualitative at this stage. This interpretation needs to be supported by additional diagnostics of deep convection (e.g., mixed-layer depth or buoyancy fluxes), which are addressed in the following section, so it may be a bit too soon to describe it here.

Figure 3a: adjust the scale to -2 → 2 so the differences appear clearer. Usually significant values are marked with dots, not the other way around. caption: since the hosing → after the hosing.

**3.2 Large-scale changes in the subpolar region**

The title may be misleading, as the AMOC is a basin-scale circulation, while this section focuses on regional changes in the subpolar North Atlantic. Consider changing it to **"3.2 Changes in the subpolar region".**

 l.164–169: The statement that changes in surface buoyancy responsible for the AMOC slowdown are "probably happening in the Labrador Sea" is plausible but not yet directly demonstrated at this stage. Consider softening the wording, e.g. "suggesting that the Labrador Sea may play an important role in the AMOC response", or explicitly framing this as a hypothesis supported later by mixed-layer depth diagnostics.

l.168–180: The attribution of freshening and cooling in the North Atlantic Current to the already developed AMOC slowdown remains qualitative. It would be helpful to briefly clarify whether these anomalies are interpreted as direct advection of freshwater from the Labrador Current or as an indirect dynamical response to reduced overturning and northward heat transport.

l.180: The acceleration of boundary currents is clearly shown but not physically interpreted. A short explanation linking this response to enhanced lateral density gradients and geostrophic adjustment associated with surface freshening would strengthen the discussion.

l.185: The lack of a significant mixed layer depth response in the Irminger and Nordic seas is an interesting result. A brief discussion of possible reasons for this regional contrast would improve the physical interpretation.

l.190–193: The absence of a clear Subpolar Gyre slowdown despite mixed layer shoaling contrasts with previous studies. A short comment on why this relationship may not hold in the present experiments (e.g. forcing magnitude or timescale) would be valuable.

l.215: The interpretation of isopycnal-aligned freshening as evidence of weak diapycnal mixing could be slightly softened. Consider rephrasing to emphasize the dominance of along-isopycnal transport rather than excluding diapycnal processes.

Figure 4: The use of March mixed layer depth is reasonable but the authors may consider also examining a winter-mean mixed layer depth (e.g. JFM) to assess the robustness of the results to the choice of month.

**3.3 Time evolution**

l.235: This robustness criterion relies again on the three-year threshold discussed above (see comment on l.120). Please ensure that the rationale for this choice is clearly stated once and applied consistently throughout the manuscript.

l.240: The weaker salinity signal in the Labrador Current is attributed to mixing with neighboring water masses. This is plausible, but remains qualitative; the wording could be softened or briefly supported by additional diagnostics.

l.245: Brine rejection is mentioned as a possible contributor to salinity changes along the Labrador Current. Please clarify whether this process is explicitly diagnosed in the model or inferred qualitatively.

l.252: Two mechanisms are proposed for the delayed freshening of the Norwegian Current. Since the delayed SPG weakening is dismissed based on the barotropic streamfunction, it would help to clarify whether this diagnostic alone is sufficient to rule out changes in salt transport.

l.259: Freshening and cooling in the NAC region are linked to the AMOC slowdown. This interpretation is reasonable, but the causal link could be phrased more cautiously, as other contributions (e.g. direct advection or air–sea fluxes) cannot be excluded.

l.270: A thermal wind response is invoked to explain the behavior at 60° N. A short clarification of why this mechanism would dominate there compared to 40–50° N would improve readability.

l.279: The temperature–salinity compensation during the first seven years is well described and likely delays the emergence of a clear density signal; stating this explicitly would help link this discussion to the delayed mixed layer depth response.

l.283: The explanation that heat anomalies are advected more efficiently than freshwater anomalies is plausible but speculative. Consider softening the wording or clearly framing this as an interpretation.

l.286: The stratification-driven shoaling of the mixed layer depth is convincing. For clarity, it may help to briefly restate that salinity dominates the density signal in the later years.

l.292–295: The comparison with other freshwater injection studies is useful and could more explicitly emphasize the role of the spatial distribution of freshwater forcing, rather than its magnitude alone.

**3.4 Global Impacts**

l.297-302: Since this paragraph summarizes results from previous studies rather than findings from the present simulations, it may be better suited for the Introduction.

l.309: Wintertime cooling over the Labrador and Nordic Seas is strong (>5 °C locally). A brief clarification on whether this reflects mean seasonal anomalies or episodic extremes amplified by air–sea coupling would improve interpretation.

l.313: Warming over the Amundsen and Ross Seas is consistent with previous studies. Given the short integration time, it would be helpful to stress that this response likely reflects atmospheric teleconnections rather than oceanic heat transport.

l.320: Interpreting the sea level pressure anomalies in terms of NAM and SAM is plausible, but remains qualitative. Please clarify whether these modes are diagnosed explicitly or inferred from spatial pattern resemblance.

l.330: Summer sea level pressure anomalies over the North Atlantic are described as a positive NAO phase: as for the NAM/SAM, it would be useful to indicate whether a formal NAO index is computed or whether this interpretation is based on pattern similarity.

l.339: Briefly contrasting this with stronger hosing experiments could further emphasize the role of AMOC amplitude.

**4. Conclusions**

l.345: Given the importance of the experimental design please briefly restate in one sentence that the simulations are performed under constant 1950 forcing and that the freshwater flux is applied in addition to the model background runoff.

l.352–355: Quantifying the AMOC reduction in density space is useful but please add values in depth space, and ensure consistency with earlier sections when citing percentage versus absolute changes.

l.364: May be useful to remind the reader that these atmospheric signals are less robust across ensemble members than the oceanic response.

 l.378: List key differences in forcing magnitude or timescale when comparing with Schiller-Weiss et al. (2024)

l.383: The statement that the initial AMOC state does not critically affect the ocean response is not fully demonstrated as there are visible inter-member differences, plus some signals become consistent only after several years plus atmospheric responses show a larger spread. Maybe "…the qualitative ocean response appears robust across different initial AMOC states, although quantitative differences remain." would be more suited.

l. 388: TipMIP → TIPMIP